# Preventive and Therapeutic Effects of *Lactiplantibacillus plantarum* HD02 and MD159 through Mast Cell Degranulation Inhibition in Mouse Models of Atopic Dermatitis

**DOI:** 10.3390/nu16173021

**Published:** 2024-09-06

**Authors:** A-Ram Kim, Seong-Gak Jeon, Hyung-Ran Kim, Heeji Hong, Yong Won Yoon, Byung-Min Lee, Chung Hoo Yoon, Soo Jin Choi, Myoung Ho Jang, Bo-Gie Yang

**Affiliations:** 1Research Institute, GI Biome Inc., Seongnam 13201, Republic of Korea; arkim@gi-biome.com (A.-R.K.); sgjeon@gi-biome.com (S.-G.J.); hrkim@gi-biome.com (H.-R.K.); heeji.hong@gi-biome.com (H.H.); 2Maeil Innovation Center, Maeil Dairies Co., Ltd., Pyeongtaek 17714, Republic of Korea; eddyoon@maeil.com (Y.W.Y.); peteia@maeil.com (B.-M.L.); milkscience@maeil.com (C.H.Y.); iamcsujin@maeil.com (S.J.C.); 3Research Institute, GI Innovation Inc., Seoul 05855, Republic of Korea

**Keywords:** allergy, mast cell degranulation, probiotics, *L. plantarum*, atopic dermatitis

## Abstract

As the relationship between the gut microbiome and allergies becomes better understood, targeted strategies to prevent and treat allergies through gut microbiome modulation are being increasingly developed. In the study presented herein, we screened various probiotics for their ability to inhibit mast cell degranulation and identified *Lactiplatibacillus plantarum* HD02 and MD159 as effective candidates. The two strains significantly attenuated vascular permeability induced by mast cell degranulation in a passive cutaneous anaphylaxis (PCA) model and, in the MC903-induced murine atopic dermatitis (AD) model, demonstrated comparable preventive effects against allergies, reducing blood levels of MCPT-1 (mast cell protease-1) and total IgE. In the house dust mite (HDM)-induced murine AD model, both *L. plantarum* HD02 and MD159 showed therapeutic effects, with *L. plantarum* HD02 demonstrating superior efficacy. Nevertheless, *L. plantarum* MD159 better suppressed transepidermal water loss (TEWL). Furthermore, *L. plantarum* HD02 and MD159 significantly increased the number of splenic Foxp3^+^ regulatory T cells, with *L. plantarum* MD159 having a more pronounced effect. However, only *L. plantarum* HD02 achieved a reduction in immune cells in the draining lymph nodes. Our findings highlight *L. plantarum* HD02 and MD159 as promising candidates for the prevention and treatment of allergies, demonstrating significant efficacy in suppressing mast cell degranulation, reducing the number of allergy biomarkers, and modulating immune responses in experimental models of AD. Their distinct mechanisms of action suggest potential complementary roles in addressing allergic diseases, underscoring their therapeutic promise in clinical applications.

## 1. Introduction

IgE produced by stimulation binds to FcεRI, a high-affinity IgE receptor overexpressed on mast cells. In the presence of a cognate antigen, cross-linking of IgE bound to FcεRI induces mast cell degranulation, triggering allergic reactions downstream [1]. During degranulation, mast cells release inflammatory mediators, including histamine and leukotrienes, leading to clinical symptoms such as itching, hives, edema, cough, and shortness of breath. In severe cases, anaphylaxis can occur, posing a life-threatening risk. Based on this understanding, suppressing mast cell degranulation has been a primary focus in allergy drug development, leading to the generation of omalizumab. Omalizumab, an anti-IgE neutralizing antibody that interferes with the binding of IgE to FcεRI, has emerged as a blockbuster drug [1,2]. Though effective, omalizumab has its limitations. The recommended dosage is determined based on blood IgE levels and body weight, requiring higher dosages for patients with elevated IgE levels or a higher body weight [3]. However, there is an upper limit for the effective dosage, and beyond established thresholds, no recommended dosage exists. Consequently, when arbitrary doses were administered to subjects in a study, atopic dermatitis patients with high blood IgE levels often showed unrelated clinical changes or even deterioration [4]. To address this issue, combination therapy involving probiotics with anti-allergy properties has recently been proposed [5]. In allergic mouse models of food allergies, IgE_TRAP_, a novel drug targeting IgE, demonstrated enhanced suppression of mast cell degranulation and allergy symptoms when administered alongside anti-allergy probiotics, without the need for dose escalation [5]. This finding underscores the potential of combining IgE-targeting therapies with anti-allergy probiotics to overcome dosage limitations and improve treatment outcomes. Therefore, the demand for probiotic strains with excellent anti-allergy efficacy is anticipated to increase.

The gut microbiome is known to be deeply involved in the development of the host immune system [6], with the authors of different studies reporting that the gut microbiome composition of children with allergies differs from that of children without allergies [7,8]. In germ-free (GF) mice, which lack a gut microbiome, B cells produce abnormally high levels of IgE through IgE isotype switching [9]. This phenomenon is CD4^+^ T cell and IL-4 dependent, indicating that IL-4-producing T helper type 2 (Th2) cells are crucial for IgE production. Consequently, GF mice exhibit more severe allergic responses than mice with gut microbiota [9,10,11]. 

Certain probiotic strains can mitigate IgE production by reducing the number of Th2 cells or increasing T helper type 1 (Th1) or regulatory CD4^+^ T (Treg) cells, thereby inhibiting IgE-induced mast cell degranulation [12]. Alternatively, some strains may directly induce mast cell apoptosis [13]. By modulating immune responses through these different mechanisms, specific probiotic strains can alleviate the symptoms of food allergies, asthma, and atopic dermatitis [14,15,16]. While the precise mechanisms of action are not fully understood, it is hypothesized that specific metabolites or proteins produced by the strains are involved [13,17,18].

In the study presented herein, we identified probiotic strains with potent anti-allergy efficacy by screening for their ability to inhibit mast cell degranulation using both in vitro and in vivo methods. The strains *L. plantarum* HD02 and MD159, recently isolated from the fermented food Kimchi and then identified, demonstrated preventive and therapeutic effects on allergy symptoms in mouse models of atopic dermatitis (AD). Notably, the mechanisms of action of these strains appeared to differ. These findings suggest that *L. plantarum* HD02 and MD159 could contribute to the prevention and treatment of allergic diseases, either alone or in combination therapy.

## 2. Materials and Methods

### 2.1. Isolation and Identification of New Probiotic Strains

Probiotics were isolated from bovine raw milk, cow feces, and fermented foods such as Kimchi and jeotgal. For each sample, 10 g was diluted with 90 mL of saline in a Whirl-Pak^®^ sampling bag (NASCO, Fort Atkinson, WI, USA) and homogenized using a BagMixer^®^ 400 VW (Interscience, Saint-Nom-la-Bretèche, France). The filtered suspension was diluted ten-fold and inoculated onto plate count agar with Bromocresol purple (BCP) (Eiken, Tokyo, Japan) using the spread plating method, and the plates were incubated at 37 °C for 48 ± 3 h. Yellowish colonies were selected and streaked twice on MRS agar plates to obtain pure strains. Each bacterial strain was identified via 16S rRNA gene sequencing using bacterial universal primers. Only bacterial strains belonging to the 19 species of lactic acid bacteria certified by the Korea Ministry of Food and Drug Safety (KMFDS) were used in the screening experiment [19].

### 2.2. Mice

Of the mice used in the experiment, the BALB/c mice were obtained from Orient Bio (Seongnam, Republic of Korea) and the NC/Nga mice were obtained from Central Lab Animal (Seoul, Republic of Korea). The mice were housed under SPF conditions and maintained in a temperature-controlled environment with a 12 h dark/light cycle. All of the animal experiments were approved by the Institutional Animal Care and Use Committee (IACUC) and performed at the GI-Biome animal facility in compliance with all regulations applicable to the management and use of laboratory animals (Approval no. PCA Model; GIB-22-02-005, MC903-Induced AD Model; GIB-22-03-001, HDM-Induced AD Model; GIB-22-03-003).

### 2.3. Mast Cell Degranulation Assay

The rat-derived mast cell line RBL-2H3 cells (American Type Culture Collection, Manassas, VA, USA) were cultured in a minimum essential medium (MEM) with Earle’s salts, supplemented with glutamine, antibiotics, and 15% fetal bovine serum (FBS). RBL-2H3 cells (1.5 × 10^5^/well) placed onto a 96-well plate were sensitized with anti-dinitrophenyl (DNP) IgE (20 ng/mL) overnight and washed twice with S-buffer containing 0.1% bovine serum albumin (BSA). The S-buffer comprises 25 mM PIPES (pH 7.2) buffer with 119 mM NaCl, 5 mM KCl, 0.4 mM MgCl_2_, 1 mM CaCl_2_, 40 mM NaOH, and 5.6 mM glucose. Thereafter, the RBL-2H3 cells were incubated with each probiotic strain for 2 h or with the cell-free culture supernatant of each strain for 3 h. The mast cells were stimulated with the antigen dinitrophenyl–human serum albumin (DNP-HSA; 25 ng/mL) for 15 min and then placed on ice to terminate stimulation. β-hexosaminidase activity was measured as an indicator of mast cell degranulation. To determine β-hexosaminidase activity, the culture supernatant or cell lysate disrupted with 0.1% triton X-100 was mixed with 1 mM p-nitrophenyl-N-acetyl-β-D-glucosaminide (p-NAG) and incubated at 37 °C for 1 h. The reaction was then stopped with 0.1 M carbonate. The color density was measured at 405 nm using a microplate reader (Molecular Devices, San Jose, CA, USA). Mast cell degranulation was quantified as the ratio of β-hexosaminidase activity in the culture media to the total activity in the culture media and cell lysate.

The live bacteria and culture supernatant of each probiotic strain for the experiment were prepared as follows. Each probiotic strain was cultured in MRS broth medium at 37 °C for 16 h under anaerobic conditions. Thereafter, the culture medium was centrifuged at 10,000× *g* and 4 °C for 5 min, and the bacterial pellet and culture supernatant were collected separately. The bacterial pellet was washed three times with phosphate-buffered saline (PBS) and then resuspended in S-buffer to achieve an OD_600nm_ value of approximately 0.8 before treatment with mast cells. When the OD_600nm_ value was 0.8, the viable cell count of each bacterial strain was experimentally measured and summarized in Appendix A. Conversely, the culture supernatant was filtered through a 0.22 μm syringe filter and diluted 1:15 in S-buffer prior to mast cell treatment.

### 2.4. Passive Cutaneous Anaphylaxis (PCA) Model

The PCA model was created as follows based on the method described in a previous report [20]. Anti-DNP IgE (20 ng per mouse) was intradermally injected into both ears of 5-week-old male BALB/c mice, and 24 h later, 100 μg of antigen DNP-HSA dissolved in Evans blue solution (5 mg/mL in PBS) was injected into the mice intravenously. After 30 min, the ears of each mouse were collected, and the dye that had leaked into the ear tissue was extracted in 700 μL of formamide at 63 °C. In addition, freeze-dried probiotic strains were prepared at a concentration of 1 × 10^10^ CFU in 200 μL of PBS per mouse and orally administered daily starting from 2 days before IgE sensitization. Dexamethasone, used as a positive control, was orally administered daily at a concentration of 10 mg/kg starting from 2 days before IgE sensitization.

### 2.5. MC903-Induced AD Model

To induce AD based on the method described in a previous report [21], 2 nmol of MC903 (Selleckchem, Houston, TX, USA) dissolved in 20 μL ethanol was applied to both ears of 8-week-old female BALB/c mice (10 μL per ear) on days 0, 5, 6, 9, 10, and 12. Ear thickness was measured with a digital caliper (Mitutoyo Corp., Tokyo, Japan). Freeze-dried probiotic strains were prepared at a concentration of 2 × 10^9^ CFU in 200 μL of PBS per mouse and orally administered daily starting from 7 days before topical application of MC903.

### 2.6. House Dust Mite (HDM)-Induced AD Model

The HDM-induced AD model was created based on the method used in a previous report as follows [22]. After shaving the dorsal hair of the 6-week-old female NC/Nga mice using an electronic clipper and hair removal cream, 150 µL of 4% SDS was applied to the dorsal skin, left to dry completely for 3 h, and then 100 mg of *Dermatophagoides farinae* body (Dfb) ointment Biostir AD (Biostir, Osaka, Japan) was applied. Dfb ointment was applied twice a week for 3 weeks to induce AD and then applied once a week for 8 weeks to maintain AD. Freeze-dried *L. plantarum* HD02 or MC159 were prepared at a concentration of 2 × 10^9^ CFU in 200 μL of PBS per mouse and orally administered daily for 8 weeks after inducing AD. Dexamethasone, used as a positive control, was applied to the dorsal skin twice a week at a concentration of 0.1%. The severity of dermatitis was evaluated once a week based on the degree of four categories of symptoms: erythema/hemorrhage, scarring/dryness, edema, and excoriation/erosion, which were scored as 0 (none), 1 (mild), 2 (moderate), and 3 (severe). The sum of the individual scores was taken as the dermatitis score. The frequency of scratching of facial or dorsal skin was counted for 10 min on day 16. Transepidermal water loss (TEWL) due to damage to skin barrier function was measured using the Tewameter TM nano (Courage & Khazaka, Köln, Germany) on day 52.

### 2.7. RNA Extraction and Quantitative RT-PCR (qRT-PCR) Analysis

Dorsal skin tissues were placed in RNA Later (Invitrogen, Waltham, MA, USA) and stored at −80 °C until further use. The stored tissues were transferred to individual tubes containing 1 mL of lysis buffer and homogenized for 10 min using a tissue homogenizer (Retsch, Haan, Germany) equipped with 5 mm stainless steel beads (Qiagen, Helden, Germany). RNA extraction was performed using an Easy-spin Total RNA Extraction Kit (iNtRON Biotechnology, Seongnam, Republic of Korea). cDNA was synthesized from the extracted RNA using a SuPrimeScript cDNA Synthesis Kit (Genetbio, Daejeon, Republic of Korea) according to the manufacturer’s protocol, using Oligo dT for cDNA synthesis from tissue RNA. The qRT-PCR was performed using SYBR green Master Mix (Bioneer, Daejeon, Republic of Korea) in an Applied Biosystems™ QuantStudio™ 3 Real-Time PCR System (Applied Biosystems, Waltham, MA, USA). The primer sequences were as follows: *Il4*, 5′-CGTCTGTAGGGCTTCCAAGG-3′ and 5′-AGGCATCGAAAAGCCCGAA-3′; *Ifng*, 5′-GAGGTCAACAACCCACAGGT-3′ and 5′-GGGACAATCTCTTCCCCACC-3′; *Tbp*, 5′-AGGGATTCAGGAAGACCACATAG-3′ and 5′-CATGCTGCCACCTGTAACTG-3′.

### 2.8. Enzyme-Linked Immunosorbent Assay (ELISA)

The total IgE level in serum was measured using ELISA MAX™ Deluxe Set Mouse IgE (BioLegend, San Diego, CA, USA) according to the manufacturer’s protocol, with serum samples diluted 1:1000. The mast cell protease-1 (MCPT-1) level in serum was quantified using a mouse MCPT-1 (mMCP-1) Uncoated ELISA Kit (Invitrogen, Waltham, MA, USA) according to the manufacturer’s protocol, with serum samples diluted 1:100.

### 2.9. Flow Cytometry Analysis

Single-cell suspensions were isolated from the lymph nodes and spleens of the mice. Prior to staining for cell surface markers, the cells were incubated with an anti-CD16/32 antibody for 10 min to block surface Fc receptors. For surface staining, the following antibodies were used: CD19 (1D3, BD Biosciences, San Jose, CA, USA), CD3 (17A2, BioLegend, San Diego, CA, USA), and CD4 (GK1.5, BioLegend, San Diego, CA, USA). The cells were then fixed and permeabilized using a Foxp3/transcription factor staining buffer set (eBioscience, San Diego, CA, USA) and stained with anti-Foxp3 antibody (FJK-16s, Invitrogen, Waltham, MA, USA). Cell analysis was conducted using FACSymphony A3 (Becton Dickinson, San Jose, CA, USA) and FlowJo_v10.10.0. software.

### 2.10. Histological Analysis

The dorsal skins were fixed in 10% neutral buffered formalin and embedded in paraffin. Tissue paraffin sections were stained with hematoxylin and eosin (H&E) to evaluate changes in epidermal layer thickness. In addition, the tissue paraffin sections were stained with toluidine blue to examine mast cell infiltration into skin lesions. The thickness of the epidermal layer and the number of mast cells were quantified using ImageJ version 1.53a software (National Institute of Health, Bethesda, MD, USA).

### 2.11. Whole-Genome Sequencing and Analysis

For whole-genome sequencing, *L. plantarum* HD02 and MD159 strains were cultured until the exponential growth phase and then harvested using centrifugation (GYLZ-1736R, Labogene, Seoul, Republic of Korea) at 8000 rpm for 5 min. The harvested cells were washed twice with PBS and then frozen until DNA extraction. Genomic DNA was extracted using a MagAttract HMW DNA kit (Qiagen, Hilden, Germany). Libraries for PacBio sequencing were prepared using the PacBio Express Template Preparation Kit 2.0 (Pacific Bioscience, Menlo Park, CA, USA); in comparison, the libraries for Illumina sequencing were prepared using the Illumina TruSeq Nano DNA Library Prep Kit (Illumina, San Diego, CA, USA). Genome sequencing with the libraries was performed using PacBio Sequel IIe and Illumina NovaSeq 6000 (DNALink, Seoul, Republic of Korea). The sequencing reads from the Illumina platform were quality-trimmed using Trimmomatic v.0.39 [23]. De novo assembly of the sequencing reads from the PacBio Sequencing platform was carried out using the microbial assembly protocol included in SMRT Link v11.0.0 and refined using the SMRT Link Arrow algorithm [24]. The refined genome sequences were validated using Pilon v.1.22 with short reads of each strain from the Illumina platform [25]. Protein-coding genes were predicted using Prodigal v.2.6.3 [26]. To estimate the genomic distances between genomes of these strains and the 296 completely sequenced genomes of *L. plantarum* in the GenBank database, the average nucleotide identity (ANI) values were calculated using the FastANI program v.1.34 [27]. The potential virulence factors were predicted through homology searches against the Virulence Factor Database (VFDB) [28].

### 2.12. Statistical Analysis

A statistical analysis of all data was performed with Prism software version 8.4.3 (GraphPad Software, Boston, MA, USA). Statistical significance was determined using regular one-way ANOVA, followed by Dunnett’s multiple comparisons test. A *p*-value < 0.05 was considered statistically significant, denoted as * *p* < 0.05, ** *p* < 0.01, and *** *p* < 0.001.

## 3. Results

### 3.1. Identification of L. plantarum HD02 and MD159 as Potent Inhibitors of Mast Cell Degranulation

To identify probiotic strains with potential anti-allergy efficacy, various probiotics isolated from bovine raw milk, cow feces, and fermented foods such as Kimchi and jeotgal were screened using an in vitro method designed to assess the inhibition of mast cell degranulation. In this experiment, the rat-derived mast cell line RBL-2H3 was used, where mast cells were sensitized with the antigen DNP-HSA in the presence of anti-DNP IgE to induce degranulation. Mast cells were treated either with live bacteria or with a cell-free culture supernatant, and the degree of inhibition of degranulation was evaluated based on β-hexosaminidase activity. As a result, mast cell degranulation was effectively inhibited by live bacteria of *Limosilactobacillus fermentum* MD58, *Lactobacillus gasseri* MK03, and *L. plantarum* HD02, MD159, and MD161 (Figure 1A). In particular, *L. plantarum* HD02 most effectively inhibited mast cell degranulation in a dose-dependent manner (Figure 1A and Appendix A). However, among these strains, all strains except *L. plantarum* HD02 and MD159 showed a limited inhibitory ability when tested with the cell-free culture supernatant. Notably, *L. plantarum* HD02 exhibited the most potent inhibitory ability in both the live bacteria and cell-free culture supernatant (Figure 1B), indicating that, unlike the other strains, it can effectively inhibit mast cell degranulation through secreted substances such as metabolites.

To validate the inhibitory effects observed in an in vivo setting, a PCA mouse experiment was conducted. This involved intradermal injection of anti-DNP IgE into the mouse ear, followed by intravenous injection of antigen DNP-HSA and Evans blue dye through the tail vein (Figure 2A). Subsequent mast cell degranulation released inflammatory mediators, increasing vascular permeability and allowing Evans blue to leak into the ear tissue. The probiotic strains *L. fermentum* MD58, *L. gasseri* MK03, and *L. plantarum* HD02, MD159, and MD161, identified as effective inhibitors of mast cell degranulation in vitro (Figure 1A), were orally administered two days before anti-DNP IgE treatment. Dexamethasone, a corticosteroid used to treat severe allergies, was used as a positive control and significantly inhibited Evans blue leakage. Notably, no Evans blue leakage was observed in the healthy mice, in which mast cell degranulation did not occur (Figure 2B,C). Among the orally administered probiotic strains, only *L. plantarum* HD02 and MD159 significantly reduced Evans blue leakage (Figure 2B,C), which were both recently isolated from the fermented food Kimchi. These results show that *L. plantarum* HD02 and MD159 effectively inhibit mast cell degranulation both in vitro and in vivo.

### 3.2. Whole-Genome Analysis of L. plantarum HD02 and MD159

Whole-genome sequencing was performed to confirm that *L. plantarum* HD02 and MD159 are different strains. As a result, *L. plantarum* HD02 and MD159 showed ANI values of 98.61% and 99.06%, respectively, for the *L. plantarum* strain DSM 20174. Of note, *L. plantarum* HD02 was genetically closest to *L. plantarum* LZ227 (ANI value = 99.44%), while *L. plantarum* MD159 was closest to *L. plantarum* SRCM103362 (ANI value = 99.76%), demonstrating that these two strains are different strains. Additionally, based on virulence factor analysis using the VFDB, these two strains each possessed 12 potential virulence genes with more than 60% identity and coverage. However, none of these genes corresponded to toxin genes but were instead genes involved in cell survival and environmental adaptation, such as GroEL, bile salt dehydrogenase, and capsular polysaccharide biosynthesis.

### 3.3. L. plantarum HD02 and MD159 Show Preventive Effects in an MC903-Induced AD Mouse Model

The preventive effects of *L. plantarum* HD02 and MD159 against allergies were investigated using a mouse model of MC903-induced AD. In this model, mice develop symptoms of acute AD such as erythema, scales, and edema in their ears following the application of MC903, a vitamin D3 derivative (Figure 3A–C). These symptoms were alleviated when *L. plantarum* HD02 and MD159 were orally administered daily starting from one week before AD induction, with both strains showing a similar degree of reduction in ear thickness (Figure 3B,C). In addition, *L. plantarum* HD02 and MD159 significantly reduced blood levels of MCPT-1, an indicator of mast cell degranulation, as well as total IgE levels, and the degree of reduction was similar (Figure 3D,E). The above results indicate that both *L. plantarum* HD02 and MD159 have a preventive effect on acute AD induced by MC903 through the suppression of mast cell degranulation and IgE production.

### 3.4. L. plantarum HD02 and MD159 Show Therapeutic Activity in an HDM-Induced AD Mouse Model

The therapeutic effect of *L. plantarum* HD02 and MD159 was investigated in a mouse model of HDM-induced AD, which closely mimics clinical AD. In this model, AD was induced by applying Dfb, an HDM, to the backs of NC/Nga mice twice a week for 3 weeks, with maintenance applications once a week for the subsequent 8 weeks (Figure 4A). *L. plantarum* HD02 and MD159 were orally administered daily for 8 weeks post induction to assess their therapeutic efficacy. Dexamethasone, a common treatment for severe allergies, served as a positive control. 

Mice with AD exhibited high dermatitis scores, with thickened skin and pronounced wrinkles from persistent scratching and rubbing due to itching (Figure 4B). *L. plantarum* HD02 and MD159 significantly reduced the dermatitis score over time, with *L. plantarum* HD02 showing an efficacy comparable to that of dexamethasone (Figure 4C). To assess itching behavior, the mice were recorded over a set period, and scratching frequency was measured. As a result, *L. plantarum* HD02 effectively reduced the scratching behavior, akin to dexamethasone, whereas *L. plantarum* MD159 did not show a significant reduction (Figure 4D). Furthermore, *L. plantarum* HD02 reduced elevated blood IgE levels; in comparison, *L. plantarum* MD159 did not show a significant reduction (Figure 4E). 

TEWL, an indicator of skin barrier function impairment, was significantly reduced by both *L. plantarum* strains and dexamethasone, with *L. plantarum* MD159 demonstrating a slightly better efficacy than *L. plantarum* HD02 (Figure 4F). Moreover, histological analysis of skin sections stained with H&E for skin thickening and toluidine blue for mast cell infiltration showed that both *L. plantarum* strains significantly reduced epidermal thickness and mast cell infiltration, comparable to the effect of dexamethasone (Figure 5).

The above results indicate that *L. plantarum* HD02 exhibits superior therapeutic efficacy against allergies compared to *L. plantarum* MD159 in the chronic AD model induced by HDM, though *L. plantarum* MD159 performs slightly better than *L. plantarum* HD02 in reducing TEWL.

### 3.5. L. plantarum HD02 and MD159 Regulate Immune Responses in an HDM-Induced AD Mouse Model

To understand how *L. plantarum* HD02 and MD159 manifest their anti-allergy effects, we examined immune responses occurring in the axillary lymph node (ALN), which serves as the draining lymph node, of mice with HDM-induced AD. The immunosuppressant dexamethasone, used as a positive control, decreased the number of total cells, T cells, and B cells that were elevated due to inflammation, as expected (Figure 6B). This suppression of the immune response was significant with *L. plantarum* HD02 but not with *L. plantarum* MD159 (Figure 6B). However, an analysis of gene expression in the ALN showed that both *L. plantarum* HD02 and MD159 significantly reduced IL-4 gene expression without significantly affecting IFN-γ gene expression (Figure 6A). Furthermore, *L. plantarum* HD02 and MD159 markedly increased the proportion of Foxp3^+^ Treg cells among T cells in the spleen, with *L. plantarum* MD159 showing a more pronounced effect compared to *L. plantarum* HD02 (Figure 6C). These findings indicate that both *L. plantarum* HD02 and MD159 are involved in regulating immune responses, albeit possibly through different mechanisms.

## 4. Discussion

In the study presented herein, we aimed to discover probiotic strains with excellent anti-allergy efficacy. We first identified strains that effectively inhibit mast cell degranulation in vitro and then tested their ability to reduce vascular permeability in a PCA mouse model (Figure 1 and Figure 2). *L. plantarum* HD02 and MD159, recently isolated from the fermented food Kimchi, exhibited excellent anti-allergy efficacy both in vitro and in vivo. In an acute AD model induced by MC903, both strains significantly reduced blood levels of the mast cell degranulation indicator MCPT-1 and total IgE, demonstrating similar allergy prevention effects (Figure 3). However, in a chronic AD model induced by HDM, which closely mimics clinical AD, *L. plantarum* HD02 showed a superior allergy treatment effect compared to *L. plantarum* MD159 (Figure 4B–D). *L. plantarum* HD02 also significantly reduced blood IgE levels; in contrast, *L. plantarum* MD159 did not (Figure 4E). In support of this fact, *L. plantarum* HD02 showed superior inhibition of mast cell degranulation compared to *L. plantarum* MD159 in all in vitro experiments using live bacteria and a cell-free culture supernatant.

Furthermore, in an HDM-induced AD model, *L. plantarum* HD02 significantly reduced *il-4* expression and the number of immune cells such as T cells and B cells in the draining lymph node; in contrast, *L. plantarum* MD159 only significantly suppressed *il-4* expression (Figure 6A,B). The ability of *L. plantarum* HD02 to efficiently suppress IgE production appears to be linked to a reduction in both B cell number and IL-4 expression, which are critical for IgE isotype switching [29]. Since IL-4 induces mast cell proliferation and is involved in the regulation of epidermal barrier integrity [30,31], *L. plantarum* HD02 appears to mitigate the increase in the number of mast cells and epidermal thickening observed in the skin by reducing IL-4 levels. *L. plantarum* MD159 was also found to significantly reduce *il-4* expression (Figure 6A) in addition to mast cell infiltration and epidermal thickening similar to *L. plantarum* HD02 (Figure 5). However, unlike *L. plantarum* HD02, *L. plantarum* MD159 did not reduce the number of B cells and blood IgE levels, resulting in less effective allergy treatment compared to *L. plantarum* HD02. Nevertheless, *L. plantarum* MD159 showed a superior ability to suppress TEWL and induce Treg cells. The above findings indicate that, while both strains have preventive and therapeutic effects on allergies, their mechanisms of action differ, warranting further research to elucidate these mechanisms. 

Among the probiotic strains tested, only *L. plantarum* HD02 and MD159 effectively inhibited the vascular permeability caused by mast cell degranulation in the PCA model (Figure 2). In a cell-free culture supernatant assay, *L. plantarum* HD02 showed the best inhibition of mast cell degranulation, followed by *L. plantarum* MD159 (Figure 1B). This finding suggests that these strains act through soluble secretions such as metabolites, rather than direct contact. In contrast, strains such as *L. fermentum* MD58, *L. gasseri* MK03, and *L. plantarum* MD161, which showed effective inhibition in the form of live bacteria but greatly reduced inhibition in the form of cell-free culture supernatants (Figure 1), likely act through direct contact with the bacteria and are less effective at influencing mast cell degranulation in distant tissues. Supporting this hypothesis, these strains did not significantly inhibit mast cell degranulation in the PCA model (Figure 2). Instead, *L. plantarum* HD02 and MD159 showed significant efficacy in both the PCA and AD models (Figure 2, Figure 3, Figure 4 and Figure 5). In particular, *L. plantarum* HD02 demonstrated superior therapeutic effects in chronic AD induced by HDM (Figure 4 and Figure 5), with results similar to that from an in vitro experiment using cell-free culture supernatants. Such observations underscore the utility of in vitro screening using a cell-free culture supernatant for identifying potent anti-allergy probiotics. Considering the need for effective suppression of allergic reactions in tissues beyond the intestines, it can be inferred that both direct contact and secretion of soluble mediators, such as metabolites, are important for achieving optimal inhibition of mast cell degranulation.

In our study, we demonstrated that *L. plantarum* HD02 and MD159 effectively alleviate allergy symptoms in mouse models of atopic dermatitis. However, it remains to be confirmed whether this efficacy is observed to a significant degree in people in different environments, with diverse dietary habits, and influenced by different genetic factors. If significant efficacy is observed in humans, these strains could be used in combination with novel allergy drugs targeting IgE to enhance drug efficacy and overcome dosage limitations, as suggested in our previous report [5]. Ultimately, these probiotic strains are expected to contribute significantly to the prevention and treatment of allergic diseases that are becoming increasingly prevalent in industrialized societies.

## 5. Conclusions

In the study presented herein, we discovered probiotic strains that effectively inhibit mast cell degranulation through in vitro assays, namely *L. plantarum* HD02 and MD159 isolated from the fermented food Kimchi. These strains alleviated vascular permeability caused by mast cell degranulation in a PCA model. Moreover, they showed preventive and therapeutic effects on allergies by inhibiting mast cell degranulation and immune responses in mouse models of AD. However, the mechanisms of action of the two strains appeared to differ.

## 6. Patents

*L. plantarum* HD02 has been patented in the Republic of Korea (2665403) and is pending PCT application (PCT/KR2023/021419). *L. plantarum* MD159 is a pending patent in the Republic of Korea (10-2023-0189458).

## Figures and Tables

**Figure 1 nutrients-16-03021-f001:**
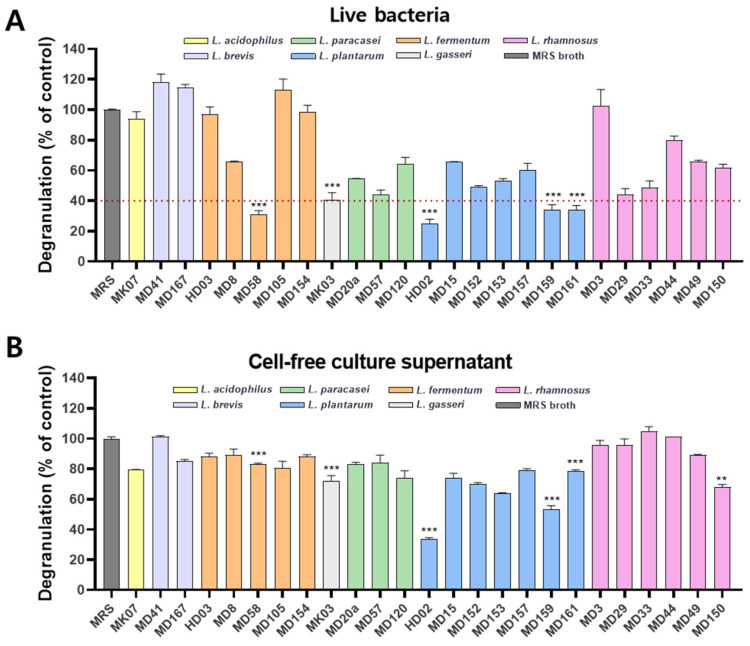
Screening of probiotic strains for the inhibition of mast cell degranulation via the β-hexosaminidase assay. Various probiotic strains were tested to identify those capable of inhibiting IgE-induced mast cell degranulation using a mast cell line RBL-2H3 and β-hexosaminidase assay. Each probiotic strain was assessed for inhibitory effects in the form of (**A**) the bacteria and (**B**) their culture supernatants. Statistical analysis was performed only on probiotic strains that inhibited mast cell degranulation by 40% or less in (**A**). Data are represented as the mean ± SD. **, *p* < 0.01 and ***, *p* < 0.001 versus the MRS group.

**Figure 2 nutrients-16-03021-f002:**
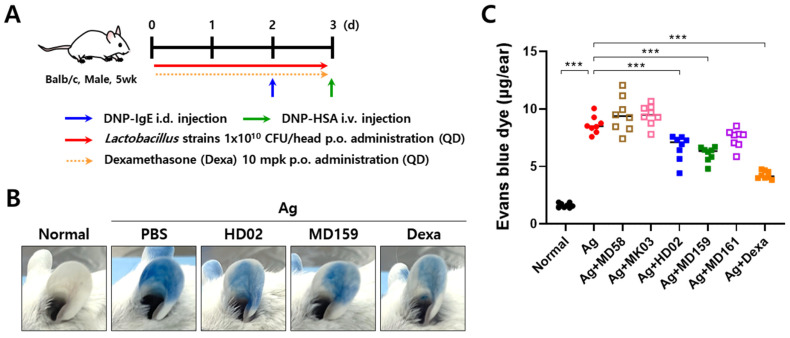
Inhibition of IgE-induced mast cell degranulation by *L. plantarum* HD02 and MD159 in a PCA mouse model. To investigate the ability of probiotic strains to inhibit mast cell degranulation in vivo, a murine model of PCA was created according to the experimental scheme displayed in (**A**). Degranulation of mast cells was measured by the extent to which Evans blue, administered through the tail vein along with the antigen DNP-HSA, leaked into the ear skin. Representative photographs are shown in (**B**), and the amount of Evans blue extracted from both ears is presented in dot plots in (**C**). i.d., intradermal; i.v., intravenous; p.o., oral; Ag, antigen; QD, once a day; *** *p* < 0.001.

**Figure 3 nutrients-16-03021-f003:**
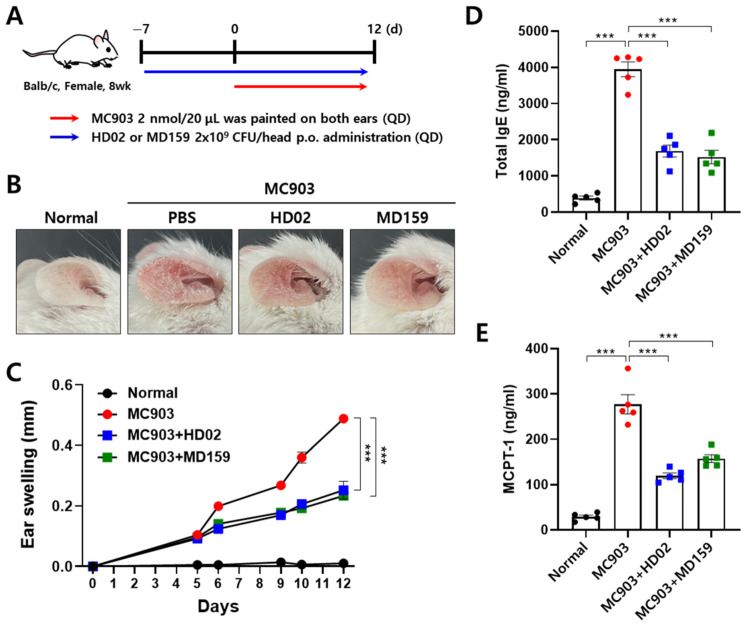
Prevention of allergic response by *L. plantarum* HD02 and MD159 in an MC-903-induced AD mouse model. To assess the preventive effects of *L. plantarum* HD02 and MD159, the mice were induced with AD based on the experimental scheme presented in (**A**). Representative images of the ear skin are shown in (**B**), and changes in ear thickness are shown graphically in (**C**). Blood levels of total IgE (**D**) and MCPT-1 (**E**) are also shown. Data are presented as the mean ± SEM. QD, once a day; p.o., oral; *** *p* < 0.001.

**Figure 4 nutrients-16-03021-f004:**
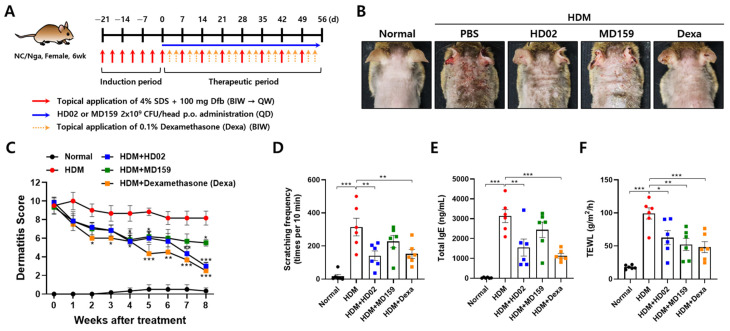
Therapeutic effects of *L. plantarum* HD02 and MD159 in an HDM-induced AD mouse model. The mice were induced with AD using HDM and treated with HD02 or MD159 according to the experimental scheme shown in (**A**). Representative images of dorsal skin are shown in (**B**), and changes in dermatitis scores are presented graphically in (**C**). The number of scratches (**D**), total IgE levels in the blood (**E**), and TEWL on the dorsal skin (**F**) were measured. Data are presented as the mean ± SEM. BIW, twice weekly; QW, once weekly; QD, once a day; p.o., oral; * *p* < 0.05; ** *p* < 0.01; *** *p* < 0.001.

**Figure 5 nutrients-16-03021-f005:**
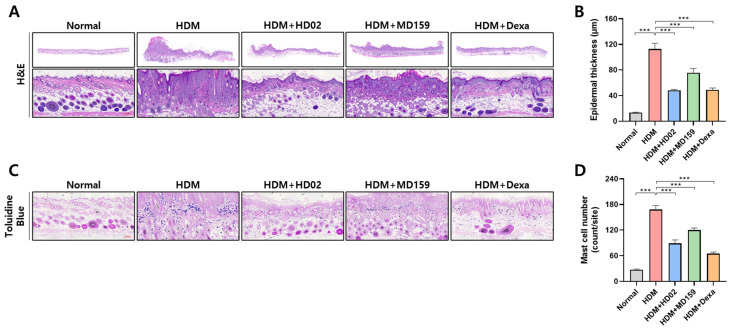
Inhibition of epidermal layer thickness and mast cell infiltration by *L. plantarum* HD02 and MD159 in an HDM-induce AD mouse model. In a mouse model of HDM-induced AD, skin legions were analyzed. Paraffin sections were stained with H&E to measure the thickness of the epidermal layer (**A**,**B**) and toluidine blue to observe mast cell infiltration (**C**,**D**). The graphs display the thickness of the epidermal layer (**B**) and the number of infiltrated mast cells (**D**). Data are presented as the mean ± SEM *** *p* < 0.001.

**Figure 6 nutrients-16-03021-f006:**
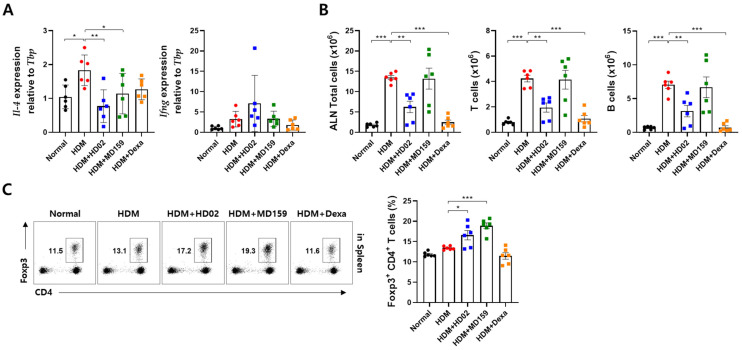
Regulation of immune responses by *L. plantarum* HD02 and MD159 in an HDM-induced AD mouse model. The involvement of *L. plantarum* HD02 and MD159 in regulating immune responses was investigated in a mouse model of HDM-induced AD by analyzing the draining lymph node, ALN, and spleen. The expressions of *il-4* and *ifng* in the ALN were examined using qRT-PCR (**A**) The numbers of total cells, T cells, and B cells in the ALN were counted (**B**). In the spleen, the ratio of Foxp3+ Treg cells among total T cells was measured via flow cytometric analysis (**C**). Data are presented as mean ± SEM. Dexa, dexamethasone; * *p* < 0.05; ** *p* < 0.01; *** *p* < 0.001.

## Data Availability

The original contributions presented in this study are included in the article; further inquiries can be directed toward the corresponding authors.

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
