# Peer review of "Preventive and Therapeutic Effects of Lactiplantibacillus plantarum HD02 and MD159 through Mast Cell Degranulation Inhibition in Mouse Models of Atopic Dermatitis"

_nutrients, 2024, doi:10.3390/nu16173021_

Round 1

Reviewer 1 Report

Comments and Suggestions for Authors

In this study, the preventive and therapeutic effects of L. plantarum HD02 and MD159 in mouse models of atopic dermatitis (AD) had been investigated. In fact, the effects of probiotic bacteria on AD have been studied before. The new aspect in this paper is the isolation of two new strains and demonstration of their effects. The results quality is fine. The following comments and suggestions can be considered:

1. Have the genomes of L. plantarum HD02 and MD159 sequenced? This is crucial to unambiguously type the two new strains.

2. Probiotic bacteria without encapsulation may have the difficulty to reach the gut. For oral administration, is it possible to measure the elevation of these two bacterial strains in the gut using molecular approach, e.g. qPCR? This is crucial for the confirmation of the mediating role of their preventive and therapeutic effects.

3. Are the effects of L. plantarum HD02 and MD159 dose-dependent? Multiple doses can better demonstrate the preventive and therapeutic effects on AD.

Author Response

For more detailed information, please refer to the attached PDF.

In this study, the preventive and therapeutic effects of L. plantarum HD02 and MD159 in mouse models of atopic dermatitis (AD) had been investigated. In fact, the effects of probiotic bacteria on AD have been studied before. The new aspect in this paper is the isolation of two new strains and demonstration of their effects. The results quality is fine. The following comments and suggestions can be considered:

  1. Have the genomes of L. plantarum HD02 and MD159 sequenced? This is crucial to unambiguously type the two new strains.

→ We have already performed whole-genome sequencing of L. plantarum HD02 and MD159 and have added the results and corresponding descriptions to the manuscript. The additions can be found in Lines 211-231 (Materials and Methods section), Lines 289-300 (Results section) and Lines 464-467 (Supplementary Materials) of the revised manuscript.

  1. Probiotic bacteria without encapsulation may have the difficulty to reach the gut. For oral administration, is it possible to measure the elevation of these two bacterial strains in the gut using molecular approach, e.g. qPCR? This is crucial for the confirmation of the mediating role of their preventive and therapeutic effects.

→ We had already demonstrated the issue mentioned by the reviewer in another paper. In the paper (Nat Comm (2022) 13:5669), we showed that B. longum has an anti-allergy effect in a food allergy model without encapsulation and can be re-cultured from the feces of oral administered mice. Of note, the BALB/c mice in the experiment were virtually free of B. longum, allowing the orally administered bacteria to be re-cultured. Additionally, according to the qPCR method, B. longum in feces which reached its peak at 4 hours after oral administration was maintained to some extent until 12 hours, and was completely eliminated at 24 hours (Figure 2). Similar results were observed in L. plantarum GB104 (Figure 1), which belongs to the L. plantarum species, suggesting that probiotics can arrive alive in the intestine without encapsulation. These results are presented only here and not in the revised manuscript because they have already been published or because L. plantarum HD02 or MD159 were not used.

Figure 1. Changes in the relative content of L. plantarum GB104 in mouse feces over time after oral administration without encapsulation.

Figure 2. Changes in fecal content over time and re-cultivation from feces of B. longum administered orally without encapsulation. These results have already been published in Nature communications [Nat comm (2022) 13:5669, Combined IgE neutralization and Bifidobacterium longum supplementation reduces the allergic response in models of food allergy].

  1. Are the effects of L. plantarum HD02 and MD159 dose-dependent? Multiple doses can better demonstrate the preventive and therapeutic effects on AD.

→ We observed that L. plantarum HD02 has dose-dependent inhibitory ability, as shown below (Figure 3). This result and the corresponding description have been added in Figure S1 and Lines 250-252 (Results section) of the revised manuscript.

Figure 3 (Supplementary figure 1). Dose-dependent inhibition of mast cell degranulation by L. plantarum HD02. The pellet of L. plantarum HD02 obtained immediately after culture was washed with PBS and then resuspended in S-buffer to prepare various doses of L. plantarum HD02 by adjusting the OD600nm values were 0.05, 0.2, 0.4, 0.6, 0.8, and 1.0, respectively. Then, each of these various doses of L. plantarum HD02 was treated to mast cells for 3 hours to investigate its inhibitory ability on mast cell degranulation. Data are represented as mean±SD.

Reviewer 2 Report

Comments and Suggestions for Authors

The manuscript by Kim A-Ram et al. submitted for review describes the anti-allergy effects of two Lactiplantibacillus plantarum strains presented as inhibition of the mast cell degranulation and in several mouse models of atopic dermatitis. I find this work rather interesting and novel, intriguing both scientifically and in terms of application. I believe that the reviewed article is of interest to the readers of “Nutrients”. However, there are several points that should be addressed:

1) L.110-111: What was the infecting dose of probiotic strains? (not in OD, but CFU/mL). Was the bacteria's toxicity tested to determine the infecting dose?

2) L.83-93: The strains were isolated from bovine raw milk, cow feces, and fermented foods and belonged to the 19 species certified by the Korea Ministry of Food and Drug Safety. I think these characteristics are not enough to recognize these strains as probiotic (throughout the manuscript).

3) Was the data in figure 1A statistically processed?

4) Expand the discussion section (discuss the major findings in the context of the literature). Are there any limitations for this study?

5) Moderate editing of English language is required.

I conclude, that the manuscript needs major revision.

Minor comments:

L.27: italicize plantarum

L.35-36: rephrase the 1st sentence

L.72: “may be” – continue the sentence

L.75: “isolated and identified from fermented food Kimchi” – isolated from fermented food Kimchi and then identified

L.91-93: “Only bacterial strains belonging to the 19 species of probiotics certified by the Korea Ministry of Food and Drug Safety were used in the screening experiment.” Add a reference (website?)

Comments on the Quality of English Language

Moderate editing of English language is required.

Author Response

For more detailed information, please refer to the attached PDF.

The manuscript by Kim A-Ram et al. submitted for review describes the anti-allergy effects of two Lactiplantibacillus plantarum strains presented as inhibition of the mast cell degranulation and in several mouse models of atopic dermatitis. I find this work rather interesting and novel, intriguing both scientifically and in terms of application. I believe that the reviewed article is of interest to the readers of “Nutrients”. However, there are several points that should be addressed:

1) L.110-111: What was the infecting dose of probiotic strains? (not in OD, but CFU/mL). Was the bacteria's toxicity tested to determine the infecting dose?

→ We have converted the OD value to CFU/mL and displayed in Lines 126-127 (Materials & methods section) of the revised manuscript.

The dose was determined at a concentration that allowed effective screening without interfering with the mast cell degranulation assay. No specific bacterial toxicity was observed during this process.

2) L.83-93: The strains were isolated from bovine raw milk, cow feces, and fermented foods and belonged to the 19 species certified by the Korea Ministry of Food and Drug Safety. I think these characteristics are not enough to recognize these strains as probiotic (throughout the manuscript).

→ To address the issue mentioned by the reviewer, we have cited the appropriate paper as references, which can be found in Line 97 (Materials and Methods section) of the revised manuscript.

3) Was the data in figure 1A statistically processed?

→ Statistical analysis have been performed according to the reviewer’s comment, which can be seen in Figure 1 and the corresponding legend in the revised manuscript.

4) Expand the discussion section (discuss the major findings in the context of the literature). Are there any limitations for this study?

→ The discussion section has been expanded based on the reviewer’s comment, which can be found in Lines 401-404 and Lines 442-450 of the revised manuscript.

5) Moderate editing of English language is required.

→ The manuscript have undergone English editing according to the reviewer’s comment.

Minor comments:

L.27: italicize plantarum

→ The typo has been corrected and can be found in Line 27.

L.35-36: rephrase the 1st sentence

→ The first sentence has been rephrased and can be found at Lines 37-38 of the revised manuscript.

L.72: “may be” – continue the sentence

→ What the reviewer said has been corrected, which can be found in Lines 75-76 of the revised manuscript.

L.75: “isolated and identified from fermented food Kimchi” – isolated from fermented food Kimchi and then identified

→ What the reviewer said has been corrected, which can be found in Lines 79-80 of the revised manuscript.

L.91-93: “Only bacterial strains belonging to the 19 species of probiotics certified by the Korea Ministry of Food and Drug Safety were used in the screening experiment.” Add a reference (website?)

→ As suggested by the reviewer, a reference has been cited, which can be found in Line 97 (Materials and Methods section) of the revised manuscript

Reviewer 3 Report

Comments and Suggestions for Authors

1.       References are missing from material and methods. Please update.

2.       Figure1. Authors should include the statistical values in figure and in legend as well along with control details.

3.       Please make the conclusion section separately.

4.       There are limitations associated with the present study, please include the separate section describing limitations of the study. 

Author Response

For more detailed information, please refer to the attached PDF.

  1. References are missing from material and methods. Please update.

→ As suggested by the reviewer, we have cited the references, which can be found in Lines 137-138, Line 148, and Lines 156-157 of the revised manuscript.

  1. Figure1. Authors should include the statistical values in figure and in legend as well along with control details.

→ As pointed out by the reviewer, we have included the results of the statistical analysis, which can be seen in Figure 1 and the corresponding legend in the revised manuscript.

  1. Please make the conclusion section separately.

→ As the reviewer noted, we have created a separate conclusion section, which can be found in Lines 451-458 of the revised manuscript.

  1. There are limitations associated with the present study, please include the separate section describing limitations of the study. 

→ What review suggested have been added, which can be found in Lines 442-445 of the revised manuscript.

Round 2

Reviewer 1 Report

Comments and Suggestions for Authors

My concerns have been appropriately addressed.

Author Response

We sincerely thank you for reviewing our manuscript.

Reviewer 2 Report

Comments and Suggestions for Authors

I thank the authors for submitting a revised manuscript and addressing my prior critiques to the best of their ability. The manuscript has been improved. Yet I do not agree with authors corrections regarding comments #1 and #2.

#1: I think, reference 20 is irrelevant. Concentration should be experimentally assessed, but not calculated (just plate cell suspension with OD 0.8 and count colonies).

#2: It is well known that probiotic properties are strain-specific. Therefore, belonging to concrete species does not guarantee that the strain will be probiotic. The anti-allergy effects shown in this paper let the authors consider SOME tested strains probiotic. Besides, in the genome sequences of L. plantarum HD02 and L. plantarum MD159 some genetic determinants of probiotic properties can be found. Throughout the document, it is necessary to check the correct use of the term “probiotic”, but not just add reference 19.

Author Response

Reviewer 2

I thank the authors for submitting a revised manuscript and addressing my prior critiques to the best of their ability. The manuscript has been improved. Yet I do not agree with authors corrections regarding comments #1 and #2.

#1: I think, reference 20 is irrelevant. Concentration should be experimentally assessed, but not calculated (just plate cell suspension with OD 0.8 and count colonies).

→ We agree with the reviewer’s comment and, as you suggested, we experimentally measured the number of each bacterial strain when cultured to an OD value of 0.8. The relevant details can be found in Lines 134-135 and Table S1 of the revised manuscript.

#2: It is well known that probiotic properties are strain-specific. Therefore, belonging to concrete species does not guarantee that the strain will be probiotic. The anti-allergy effects shown in this paper let the authors consider SOME tested strains probiotic. Besides, in the genome sequences of L. plantarum HD02 and L. plantarum MD159 some genetic determinants of probiotic properties can be found. Throughout the document, it is necessary to check the correct use of the term “probiotic”, but not just add reference 19.

→ We fully understand the reviewer’s point and have replaced “probiotics” with “lactic acid bacteria”. This can be confirmed at Line 98 of the revised manuscript.
